# Quantitative MODS-Wayne assay for rapid detection of pyrazinamide resistance in *Mycobacterium tuberculosis* from sputum samples

Emily Toscano-Guerra,[1] Roberto Alcántara,[2] Katherine Lozano Untiveros,[1] Robert Gilman,[3] Louis Grandjean,[4] Mirko Zimic,[1] Patricia Sheen[1]

**ABSTRACT**    Tuberculosis (TB) remains a significant global health challenge, exacerbated by the emergence of drug-resistant strains, such as those resistant to pyrazinamide (PZA). The current scarcity of affordable and precise quantitative diagnostic tests for PZA resistance underscores the urgent need for more accessible diagnostic tools. We evaluated PZA susceptibility in 264 TB-positive samples by quantifying pyrazinoic acid (POA) production, using both the MODS-Wayne qualitative assay and our newly developed quantitative approach (MODS-WQ). The MODS-WQ was assessed in 7H9 medium (MODS-WQ$_{7H9}$) or citrate buffer (MODS-WQ$_{CB}$), with POA levels measured via spectrophotometry against a calibration curve. PZA susceptibility determinations were based on a composite reference standard. Associations between POA levels and pyrazinamidase mutations were explored. The composite standard detected PZA resistance in 23.5% of the samples, which accounts for 62.8% of the multidrug-resistant (MDR) samples. The MODS-WQ established specific POA cutoffs of 123.25 µM for MODS-WQ$_{7H9}$ and 664.7 µM for MODS-WQ$_{CB}$, with sensitivities of 81.3% and 92.3% and specificities of 77.2% and 95.9%, respectively. Notably, samples with mutations in the pyrazinamidase metal-binding site exhibited significantly lower POA levels compared with mutations in the enzyme periphery. Furthermore, a significant correlation was found between POA production and PZA resistance, Bactec Growth Index, and minimum inhibitory concentration (MIC) values. This study presents a novel, direct, and accessible susceptibility test for PZA resistance that quantifies POA, enhancing the detection capabilities for this condition. The citrate-buffered MODS-WQ assay demonstrated high sensitivity and specificity for quantifying POA, confirming that POA production is a reliable indicator of PZA resistance.

**IMPORTANCE**    PZA susceptibility testing continues to be a challenge, particularly in countries with high TB incidence. In response to this pressing need, we have developed a quantitative MODS-Wayne (MODS-WQ) assay. This approach offers a direct and cost-effective solution representing a significant advancement in TB diagnostics, particularly benefiting resource-limited laboratories, primarily in developing regions. The MODS-WQ assay stands out for its ability to quantify pyrazinoic acid (POA) production, as a reliable indicator of PZA resistance. Unlike traditional qualitative assays, MODS-WQ eliminates the inherent subjectivity in interpretation, providing more accurate and actionable results. Moreover, the MODS-WQ approach accounts for critical factors influencing PZA resistance, including enzymatic efficiency and efflux pump activity. By integrating these factors into the detection process, our methodology offers a comprehensive understanding of PZA resistance levels, enabling tailored treatment strategies for patients.

**KEYWORDS**    Tuberculosis, pyrazinamide, pyrazinoic acid, resistance, MODS-Wayne

**Peer Reviewer** Xia Yu, Beijing Tuberculosis & Thoracic Tumor Research Institute, Beijing, China

Address correspondence to Emily Toscano-Guerra, emily.toscano.g@upch.pe, or Patricia Sheen, patricia.sheen@upch.pe.

The authors declare no conflict of interest.

See the funding table on p. 13.

Tuberculosis (TB) is still a major global health concern, particularly in low- and mid-income countries, where it has reached pandemic proportions. The emergence of drug-resistant TB strains poses a significant challenge to eradication efforts (1). Pyrazinamide (PZA), a first-line anti-TB drug, is pivotal for shortening the treatment duration to 6 months for both drug-sensitive and multidrug-resistant (MDR) TB cases (2). PZA acts as a prodrug and is converted into its active form, pyrazinoic acid (POA), by the enzyme pyrazinamidase (PZase). The PZase enzyme is encoded by the *pncA* gene and plays a crucial role in drug efficacy (3).

Resistance to pyrazinamide (PZAr) significantly worsens TB treatment outcomes and increases mortality rates (4). PZAr is estimated to occur in approximately 16.2% of all TB cases globally (5) and constitutes approximately 60% of MDR TB cases. The lack of a standardized and reliable method for routinely detecting PZAr presents a major obstacle to TB management strategies (6).

Routine testing for PZAr is often not performed because of various technical challenges (7–9). Existing phenotypic culture-based drug susceptibility testing (pDST) methods, such as the BACTEC MGIT 960 system (10), conventional Wayne assay (11), nitrate reductase assay (12), and Microscopic Observation Drug Susceptibility (MODS) assay (13), have several limitations. These include difficulties in maintaining the precise inoculum amount and pH level (5.5–6.5) necessary for PZA activity, the risk of contamination with other bacterial strains leading to an error rate of 10%–15% (9), and prolonged detection times (9). Additionally, despite its prohibitive cost, the BACTEC MGIT 960 system produces a significant rate of false resistance (8).

The conventional Wayne test, which assesses the enzymatic activity of pyrazinamidase (PZase), is an economical colorimetric test that detects pyrazinoic acid (POA) as a marker of susceptibility using ferrous ammonium sulfate (FAS) (11). Despite its cost-effectiveness, this assay has limitations: it is qualitative and subjective, and it requires extended incubation periods until bacterial growth is detectable (7).

Genotypic drug susceptibility testing (gDST) methods, such as Sanger sequencing and whole-genome sequencing, have been employed to identify specific genetic mutations in the *pncA* gene linked to PZAr (14, 15). These methods can also detect mutations in other genes including *rpsA*, *panD*, and *clpC1* (16–19). gDSTs are generally more sensitive and specific than phenotypic methods, with the majority (70%–97%) of PZA-resistant clinical isolates showing genetic variations in the promoter region of *pncA* or the coding sequence (20, 21). However, these techniques require significant technical expertise and are costly, making them less accessible in low- and mid-income countries where TB is endemic (22). Additionally, the variability in mutation patterns linked to PZAr can differ regionally, affecting the representativeness of these tests (23).

The MODS assay is an affordable and efficient test utilizing liquid culture to diagnose TB and determine drug resistance directly from sputum samples within 5–21 days (13, 24, 25). It involves the observation of the characteristic cords of *Mycobacterium Tuberculosis* (MTB) grown using inverted microscopy, which has been adapted to different low-cost versions (26). MODS is rooted in the principle that MTB grows faster in liquid than in solid medium. Although the Löwenstein-Jensen solid culture method is simple and cost-effective, it requires lengthy culture periods (6–8 weeks) for DST determination (27). The MODS assay has proven to have significant advantages over the standard BACTEC MGIT for DST determination, being performed directly from sputum with a shorter turnaround time (eliminating the need for primary MTB isolation in MGIT culture). In addition, the MODS assay has been extended to implement DST for first- and second-line drugs (28, 29). However, its application in PZA testing has been limited. To address this issue, Alcántara et al. (30) developed the MODS-Wayne method by combining the MODS assay with the Wayne test. This adaptation is based on the detection of pyrazinoic acid (POA), the hydrolyzed product of PZA expelled into the extracellular environment via a colorimetric reaction with ferrous ammonium sulfate (FAS) (11). The MODS-Wayne assay showed a high sensitivity and specificity (92.7% and 99.3%, respectively). Nonetheless, its subjective nature may have affected the reproducibility of the results.

In this study, we aimed to refine the MODS-Wayne test by introducing a quantitative approach (MODS-WQ) using sputum samples from patients with TB. This new approach seeks to reduce the diagnosis time, eliminate the need for pre-isolation in solid cultures, and move beyond qualitative judgments.

## MATERIALS AND METHODS

### Samples and study design

A total of 264 TB-positive anonymous sputum samples from our biobank were evaluated in the present study. The samples were collected from two medical centers (Hospital Nacional Dos de Mayo and the Regional Tuberculosis Reference Laboratory, Callao, Lima, Peru), during 2015–2016 (samples coded as MP) and 2017–2018 (samples coded as TBN). All samples were evaluated by the acid-fast smear (AFS) testing positive for *M. tuberculosis*. This study received approval from the Institutional Committee on Research Ethics (CIEI) of the Universidad Peruana Cayetano Heredia (Code 331–34-21). Only basic demographic data including age, sex, and treatment were recorded, and no informed consent was required for sample anonymity.

Samples were subjected to analysis using the MODS assay to assess resistance to isoniazid and rifampicin (MODS-MDR) and the qualitative MODS-Wayne assay to evaluate pyrazinamide resistance. Furthermore, a quantitative variant of the MODS-Wayne assay (MODS-WQ) was implemented, encompassing two versions utilizing distinct buffer matrices. The first version, conducted in a citrate buffer matrix with a pH of 7.0, was designated MODS-WQ$_{CB}$ ($N = 169$). The second version, performed in 7H9 broth with a pH of 6.8, was identified as MODS-WQ$_{7H9}$ ($N = 95$) (Fig. 1).

### Multidrug resistant test: rifampicin (RIF) and isoniazid (INH) resistance

All sputum samples were pre-decontaminated following the Kent and Kubica protocol (31) and inoculated in Middlebrook 7H9 medium (Supplementary Methods). The MODS test was conducted using the MODS-MDR protocol established in our laboratory (MODS-MDR) (32). Samples were incubated with isoniazid (INH) at a final concentration of 0.4 µg/mL or rifampicin (RIF) at 1.0 µg/mL (10) at 37°C for a maximum of 21 days, followed by microscopic evaluation using an inverted microscope.

### Qualitative MODS-Wayne assay for PZA susceptibility

This assay was conducted simultaneously with the MODS-MDR test (Fig. S1A). The samples were distributed into two distinct wells: the control well (PZA-C) and the treatment well (PZA-Wayne), and the plate was incubated at 37°C. Upon observing growth in the PZA-C well by microscopic evaluation, the date was documented, and incubation continued for an additional 3 days. Subsequently, PZA was added to the PZA-Wayne wells at a final concentration of 800 µg/mL and further incubated for 3 days. After incubation, 100 µL of 10% ferrous ammonium sulfate (FAS) was added and incubated for 5 min, followed by immediate color recording. To classify the test outcome, a qualitative color intensity scale was used, with values ranging from 0 (absence of POA production) to 3 (high POA production). The absence of POA was associated with PZA resistance, whereas values from 1 to 3 were associated with PZA susceptibility (Fig. S1C and D). For more details, see the Supplementary Methods.

### Quantitative MODS-Wayne assay for PZA susceptibility

This assay encompasses two variants, MODS-WQ$_{CB}$ (10 mM citrate, pH 7.0) and MODS-WQ$_{7H9}$ (pH 6.8), conducted in separate 24-well plates (Fig. S1B). The samples were incubated as described for the MODS-Wayne assay, following a similar initial incubation step. After observing positive growth in the PZA-C wells on the third day, the samples were subjected to variant-specific treatments. For MODS-WQ$_{7H9}$, sample volumes were

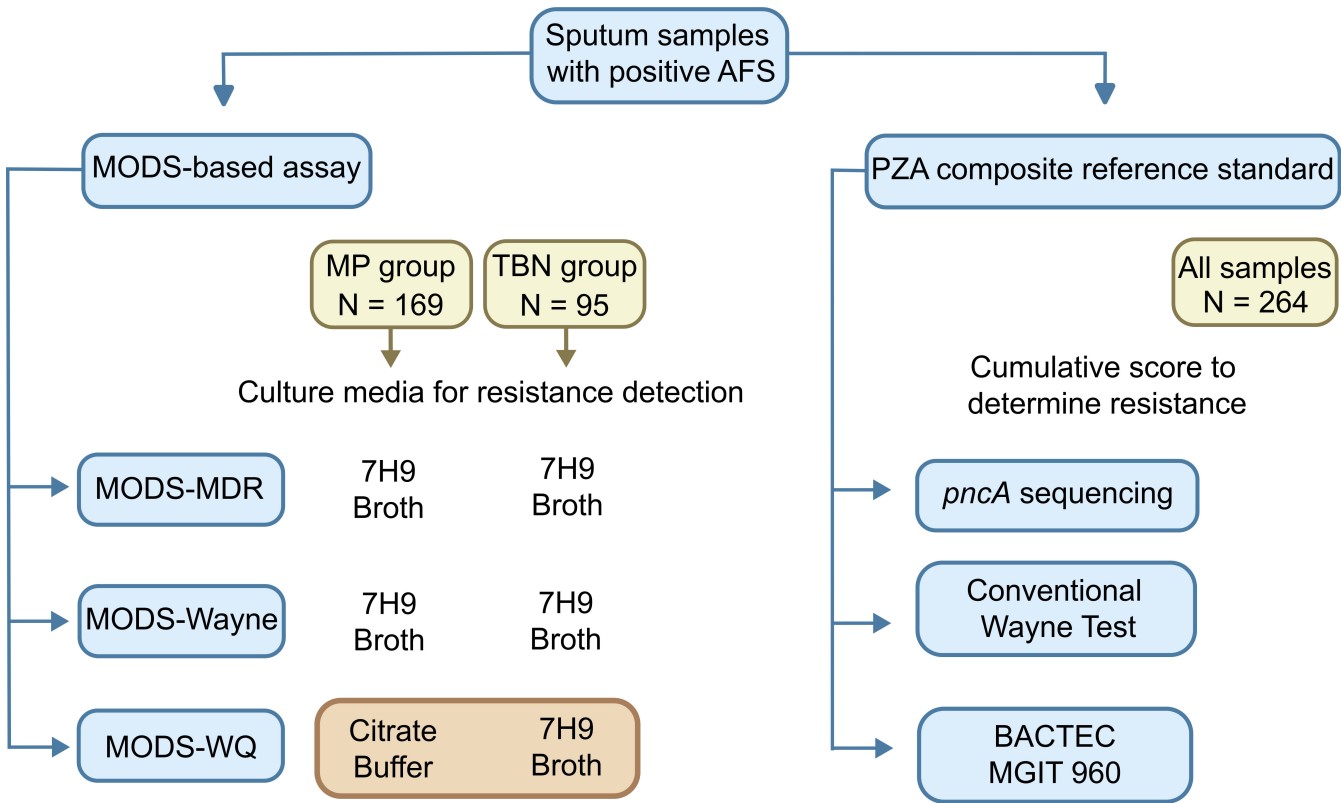

**FIG 1** Overview of study design and experimental approach. Only samples testing positive for acid-fast smear were selected for further analysis. These samples underwent evaluation through four distinct experimental assays: MODS-MDR for tuberculosis multidrug-resistant characterization, MODS-Wayne for qualitative assessment of pyrazinamide (PZA) resistance, MODS-WQ as the quantitative variant of the assay, and the composite reference standard derived from the conventional Wayne test, *pncA* gene sequencing, and the BACTEC MGIT960 PZA susceptibility testing. The diagnostic accuracy of both the MODS-Wayne and MODS-WQ assays was determined in comparison to the composite reference standard.

adjusted to 1 mL by adding 100 µL of 7H9 media to the PZA-C well and 100 µL of PZA (final concentration of 800 µg/mL) to the PZA-WQ well, followed by an additional incubation of 3 days. In contrast, for the MODS-WQ$_{CB}$ approach, samples were collected from plate wells and centrifuged. The resulting cell pellet was resuspended in 900 µL of citrate buffer and transferred to a 24-well plate. Then, 100 µL of citrate buffer was added to the PZA-C well, and 100 µL of PZA (final concentration of 800 µg/mL) to the PZA-WQ well, with further incubation for 3 days. To determine POA production, for the MODS-WQ$_{CB}$ approach, 100 µL of 10% FAS was added, and the mixture was incubated for 5 min. Subsequently, 500 µL was transferred to cryotubes and stored at −80°C for further analysis.

Conversely, for MODS-WQ$_{7H9}$ samples, centrifugation (13,000 rpm for 2 min) was performed, and only the supernatants were analyzed. For POA determination by absorbance measurements, triplicate samples (100 µL each) were transferred to a 96-well plate and measured at a wavelength of 450 nm using a spectrophotometer. Absorbance values for both approaches were then interpolated against their respective standard pyrazinoic acid (POA) curves, ranging from 31.25 µM to 4,000 µM (Table S1). For more details, see the Supplementary Methods.

## Composite reference standard for PZA susceptibility determination

In the absence of a universally acknowledged standard test, we employed a composite reference standard comprising three distinct assays: (i) BACTEC MGIT960 system, (ii) conventional Wayne test, and (iii) sequencing of the *pncA* gene (Fig. S2A). To determine the PZA susceptibility profile, a score of 0 or 1 was assigned when a susceptible or

resistant result was observed in all three assays. Finally, the samples were classified as PZA-resistant when a cumulative score of 2 or 3 was obtained from the three assays performed, or PZA susceptible to a cumulative score of 0 or 1 (Fig. S2B). Mutations were categorized based on the WHO catalog (33) and SuspectPZA webtool (https://biosig.lab.uq.edu.au/suspect_pza/). The performances of both the qualitative MODS-Wayne and MODS-WQ assays were evaluated in comparison to this composite reference standard. The detailed methodology is provided in the Supplementary Methods.

## Assessment of pyrazinamide susceptibility using MIC and BACTEC Growth Index

The susceptibility to PZA was assessed through the determination of the MIC using 7H9 culture medium (PZA-MIC) and the Bactec Growth Index (BGI), as reported in previous studies (34). These assays were only performed for isolates with the following mutation in the *pncA* gene: K48T, Q10P, Q10R, D49N, and H51R. In brief, 7H9 culture broth at a pH of 6.0 was employed to ascertain the PZA-MIC. The BGI, recognized as the gold standard for predicting PZA resistance, was used to estimate the resistance level to PZA, quantifying the radioactive growth index as a percentage.

## Impact of PZase mutations on pyrazinoic acid production

To evaluate the effect of PZase mutations on the production of pyrazinoic acid (POA), we analyzed POA concentrations (µM) across samples, correlating them to the mutation sites. These sites were categorized into several functional areas: the active catalytic site (ACS), metal binding site (MBS), PZA binding site (PBS), enzymatic core (EC), and the enzyme periphery (PER). Mutations were analyzed using PyMOL version 2.5.4, with an educational license. The 3D structure of PZase was downloaded from Protein Data Bank, entry ID: 3PL1 https://doi.org/10.2210/pdb3PL1/pdb

## Statistical analysis

Continuous variables were presented as median ± 95% CI (CI) or median and inter-quartile range (IQR). A comparative analysis of the samples was conducted using the non-parametric Mann-Whitney U test. The sensitivity and specificity of the qualitative MODS-Wayne assay were determined by using a 2 × 2 contingency table. The predictive accuracy of the MODS-WQ assay was assessed using logistic regression modeling, enabling the calculation of the receiver operating characteristic (ROC) curve, along with sensitivity and specificity values. The concordance between the assays and composite reference standard was quantified using the kappa coefficient. Linear regression analysis was used to test the association between POA production and PZA susceptibility. Statistical analyses were performed using STATA version 14 and GraphPad Prism version 9.0.2.

## RESULTS

### Study population and MDR characterization

The demographic and clinical characteristics of the patients are summarized in Table 1. There were no significant differences in the age or sex distribution between the MP and TBN groups. Notably, only 11.00% of patients in the MP group had received prior treatment compared with 67.00% in the TBN group. Of the 264 samples collected, 78 (29.54%) were identified as MDR. Among the non-MDR samples ($n = 186$, 70.45%), monoresistance was observed in 26 (9.8%) to INH, 15 (5.7%) to RIF, and 145 (55.30 %) samples that were susceptible to INH and RIF.

### Composite reference standard tests for PZA susceptibility determination

A PZA-susceptibility profile was established according to a composite standard, which was based on the result of each individual assay (Table S2). As a result, 79 (83.2 %)

**TABLE 1** Demographic information and tuberculosis multidrug-resistant profile of samples[a]

| | MP | TBN | Total | P-value[b] |
|---|---|---|---|---|
| | N = 169 | N = 95 | N = 264 | |
| Age (years), median (IQR) | 36 (27.5) | 29 (21) | 33 (24.5) | 0.3118 |
| Sex, N (%) | | | | |
| Males | 96 (57.0) | 66 (69.0) | 162 (61.4) | 0.0486[c] |
| Females | 54 (32) | 27 (31) | 81 (30.7) | 0.5795[c] |
| N.D. | 19 (11) | 2 (2) | 21 (7.8) | 0.0081 |
| Treatment, N (%) | | | | |
| No treated | 83 (49.1) | 29 (30.5) | 112 (42.4) | 0.0055[c] |
| Treated | 18 (10.7) | 64 (67.4) | 82 (31.1) | 0.0001[c] |
| N.D. | 68 (40.2) | 2 (2.1) | 70 (26.5) | 0.0042 |
| MDR | 52 (30.8) | 26 (27.4) | 78 (29.5) | |
| Mono-resistant | | | | |
| Rifampicin | 15 (8.9) | 0 (0.0) | 15 (5.7) | 0.0014 |
| Isoniazid | 20 (11.8) | 6 (6.3) | 26 (9.9) | 0.1971 |
| Sensitive | 82 (48.5) | 63 (66.3) | 145 (54.9) | 0.0067 |

[a]MP, samples collected in the study 2015-2016; TBN: samples collected in the study 2017-2018; IQR, interquartile range; MDR, multidrug resistance; N.D., no determined.
[b]P-value of difference between MP and TBN groups
[c]Analysis including N.D samples.

isolates were sensitive to PZA and 16 (16.8 %) resistant in the TBN group ($n$ = 95), whereas in the MP group ($n$ = 169), 123 (72.8 %) isolates were sensitive and 46 were resistant (27.2%) (Table S3). Regarding mutations in PZase, *pncA* sequencing revealed 187 isolates with wild-type characteristics (TBN: 71, MP: 116) and 77 isolates with mutations linked to PZA resistance, including eight undetermined cases (Table S2). Among these mutations, 16 were point mutations, two deletions, and one insertion, as outlined in Table 2. Notably, the mutation distribution was as follows: one (1.5%) in the promoter site, one (1.5%) in the catalytic site, four (5.8%) in the metal-binding site, five (7.3%) in the periphery, and six (8.7%) in the enzyme core (Table 2).

## MODS-Wayne qualitative assay performance

In the MP group, 24.26% of the samples were classified as PZA-resistant, whereas 75.74% were classified as susceptible, based on the defined color scale criteria. In contrast, these proportions were discernibly different in the TBN group, with 15.79% classified as resistant and 84.21% as susceptible (Table S4). Comparing these results with the composite reference standard, the MODS-Wayne assay demonstrated the best results in the MP group, showing a sensitivity of 87.8% (95% CI: 0.7446–0.9468, Wilson-Brown test) and a specificity of 95.9% (95% CI: 0.9084–0.9825), with a kappa index of 0.83 ($P$ < 0.0001) (Table 3). Since the samples used with this methodology had the same matrix (medium 7H9), we decided to conduct a more detailed analysis of the performance of this methodology by combining the two groups (MP and TBN). The results according to the composite standard showed a sensitivity of 80.70% (95% CI: 0.6866–0.8887) and a specificity of 95.1% (0.9113–0.9723), a $P$ value < 0.0001 and a kappa index = 0.76 (Table 3).

## MODS-WQ assay performance

The samples were classified as susceptible or resistant based on the composite reference standard outcome. For the MODS-WQ$_{7H9}$ samples, the median POA production, as determined by the calibration curve (Fig. 2A), was 274 µM (IQR = 373.3 µM) for the susceptible isolates, in contrast to 92 µM (IQR = 77.95 µM) for the resistant isolates (Fig. 2B). The Mann-Whitney test indicated a significant p-value (<0.0001), although no pronounced discrimination was observed. In contrast, MODS-WQ$_{CB}$ samples showed a marked difference in the level of POA production ($P$ < 0.0001). The median concentration

**TABLE 2** Classification and analysis of mutations detected in this study[d]

| Mutation | N° samples | Location in the PZase | Susceptibility phenotype | | PZA susceptibility profile[c] |
|---|---|---|---|---|---|
| | | | WHO[a] | SuspectPZA[b] | |
| I6S | 2 | EC | Uncertain | R | R |
| D8E | 4 | ACS | A w RI | R | R |
| Q10P | 2 | EC | A w R | R | R |
| Q10R | 14 | EC | A w R | R | R |
| A30T | 1 | PER | - | S | S |
| K48T | 9 | EC | Uncertain | R | R |
| D49N | 3 | MBS | A w RI | R | R |
| H51R | 12 | MBS | A w R | R | R |
| H57L | 3 | MBS | - | R | R |
| H57R | 1 | MBS | A w R | R | R |
| P62S | 1 | PER | A w RI | R | R |
| H71R | 2 | EC | A w R | R | R |
| F81S | 7 | PER | Uncertain | R | R |
| H82D | 1 | PER | Uncertain | R | R |
| V180F | 3 | PER | A w R | R | R |
| Prom A-11G | 1 | Promoter | A w R | N/A | R |
| 407insA/ 461insA | 1 | EC | A w RI | N/A | R |
| Δ456–466 | 3 | PER | A w RI | N/A | R |
| Δ375–389 | 1 | PER | A w R | N/A | R |
| WT | 187 | - | S | S | S |

[a]Mutation classification according to the WHO catalog.
[b]Mutation classification according to the SuspectPZA web tool. Promoter region, deletions and insertions are not considered in the tool.
[c]Mutation classification considering a and b.
[d]R, Resistant; S, sensitive; Uncertain, Ambiguous association, A w R, associated with resistance; A w RI, associated with intermediate resistance; N/A, Not Applicable

for the susceptible group was 2105 µM (IQR = 1672 µM), whereas that for the resistant group was 258 µM (IQR = 1383.3 µM) (Fig. 2B).

ROC curve analysis showed good performance for both variants. However, the area under the curve (AUC) for MODS-WQ$_{BC}$ was 97.57% (95% CI: 95.30%–99.84%), significantly higher (P-value = 0.0037) than that for MODS-WQ$_{7H9}$ (AUC = 82.28%, 95% CI: 71.28%–93.28%) (Fig. 2C), indicating a superior discriminatory capacity for MODS-WQ$_{CB}$. The optimal cutoff points derived from the ROC curves were 126.70 µM for MODS-WQ$_{7H9}$, correlating with a sensitivity of 81.25% (95% CI: 56.99%–93.41%) and a specificity of 77.22% (95% CI: 66.83%–85.07%) (Table 3 Fig. 2D). Conversely, for MODS-WQ$_{CB}$, the optimal cutoff was 664.7 µM, demonstrating a sensitivity of 92.31% (95% CI: 79.68%–97.35%) and a specificity of 95.93% (95% CI: 90.84%–98.25%) (Table 3; Fig. 2D). The kappa index resulted in 0.83 (P-value = 0.001) for MODS-WQ$_{7H9}$ and 0.87 (P-value = 0.001) for MODS-WQ$_{BC}$.

**TABLE 3** Comparative performance between qualitative MODS-Wayne and quantitative MODS-Wayne (MODS-WQ) variants[c]

| Assay | Sample group | Frequency (n/N) | Sensitivity [95% CI] | Frequency (n/N) | Specificity [95% CI] | PPV | NPV | Kappa |
|---|---|---|---|---|---|---|---|---|
| MODS-Wayne | MP | 36/41 | 0.878 [0.745–0.947] | 118/123 | 0.959 [0.901–0.983] | 0.88 | 0.95 | 0.83 |
| | TBN | 10/16 | 0.625 [0.387–0.815] | 74/79 | 0.937 [0.860–0.973] | 0.67 | 0.93 | 0.57 |
| | MP + TBN | 46/57 | 0.807 [0.686–0.889] | 192/202 | 0.951 [0.911–0.973] | 0.82 | 0.95 | 0.76 |
| MODS-WQ$_{BC}$[a] | MP | 38/41 | 0.923 [0.797–0.974] | 118/123 | 0.959 [0.908–0.983] | 0.88 | 0.98 | 0.87 |
| MODS-WQ$_{7H9}$[b] | TBN | 13/16 | 0.813 [0.569–0.934] | 61/79 | 0.772 [0.668–0.851] | 0.42 | 0.95 | 0.43 |

[a]Cutoff point of 665 µM.
[b]Cutoff point of 127 µM.
[c]Measures of accuracy (sensitivity, specificity, positive predictive value PPV, negative predictive value NPV) and confidence intervals were observed for each type of test evaluated.

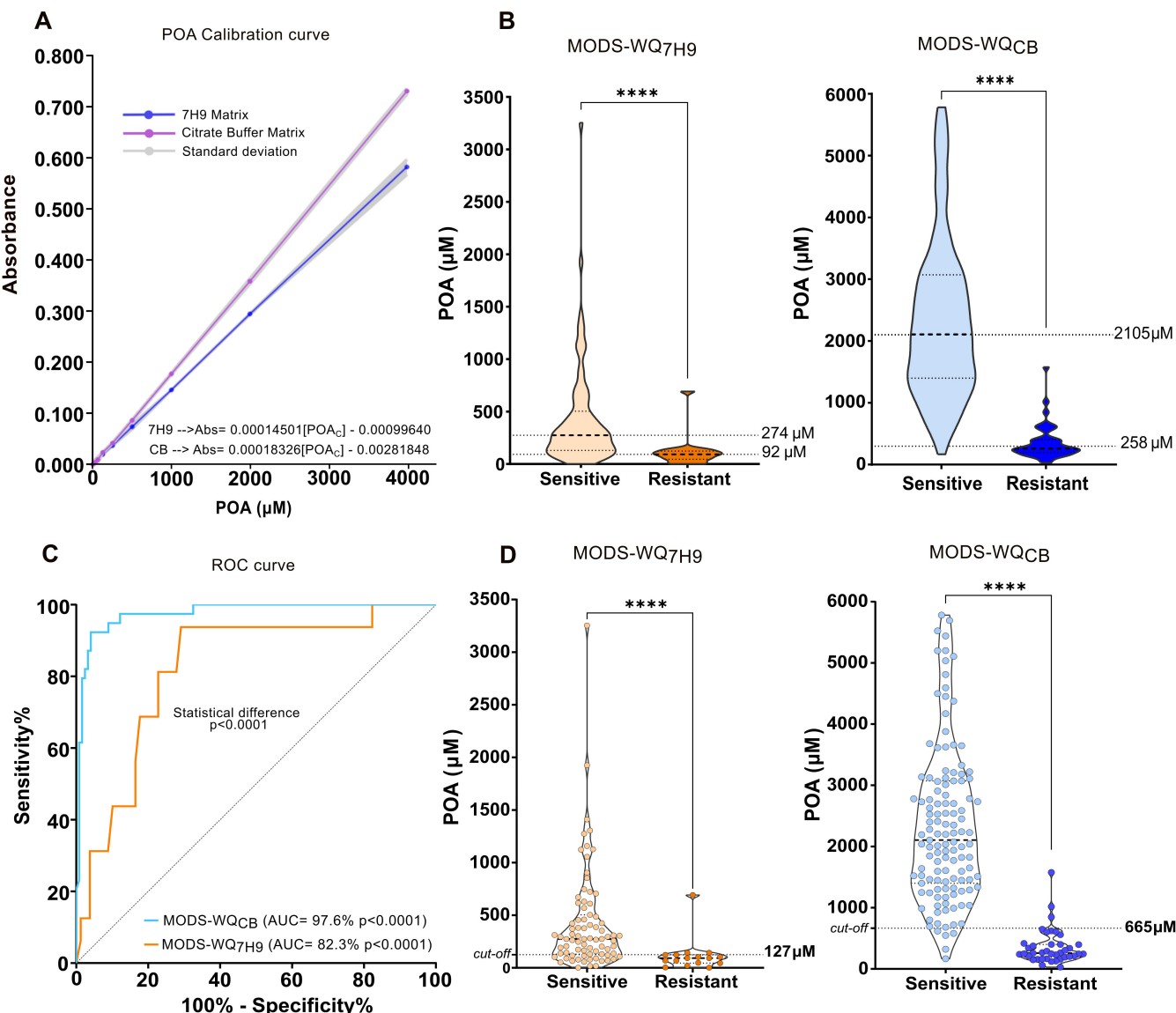

**FIG 2** Evaluation of MODS-WQ variants' performance. (A) Standard curves for pyrazinoic acid (POA) in two distinct buffer matrices, based on three replicates and their standard deviations. The formula for calculating POA concentration (POAc) is also presented. (B) Violin plot illustrating the distribution of sample classifications according to the composite reference standard, alongside measured POA concentrations in µM for both MODS-WQ7H9 and MODS-WQCB variants. Median POA concentrations for each group are indicated. (C) ROC curves and corresponding *P*-values for MODS-WQCB and MODS-WQ7H9 variants highlight statistical differences between them. (D) Violin plot demonstrating sample classifications per MODS-WQ7H9 and MODS-WQCB, including the established cutoff point for each. Statistical significance is denoted by asterisks, with * indicating $P < 0.05$, **$P < 0.01$, ***$P < 0.001$, and ****$P < 0.0001$.

## Impact of PZase mutations on pyrazinoic acid production

We analyzed POA production (µM) by PZA-resistant and -susceptible mutations, both individually and by methodology (MODS-WQBC or MODS-WQ7H9) (Table S5). Notably, the sensitive mutation A30T in MODS-WQ7H9 samples exceeded the established cutoff (126.70 µM). Furthermore, the K48T mutation, classified as "uncertain" by the WHO, showed POA values close to the method-specific cutoff points and exhibited a large IQR. Overall, we observed that most resistant mutations had a wide IQR, suggesting that factors beyond PZase activity may influence the final amount of POA production.

Although analyzing POA production for each mutation individually provides valuable insights, we were particularly interested in understanding how mutations impact POA production when considered within specific enzyme regions. Evaluating POA production

by regions, rather than solely focusing on individual mutations, offers a more comprehensive understanding of how mutations in these areas collectively impact POA production and PZA resistance. Therefore, we analyzed POA production across different enzyme regions in isolates with mutations, excluding those with deletions or promoter-site mutations (Fig. 3A). As the MODS-WQ$_{BC}$ method showed the best performance, this analysis was conducted using these POA values.

Our results indicate that most mutations, regardless of their location, result in POA production below 665 µM (cutoff point), suggesting that although mutations affect enzymatic activity, the impact is not entirely drastic as POA production persists.

Interestingly, mutations in the periphery (PER) (e.g., P62S and F81S) showed a median POA production of 644.3 µM, which is close to the cutoff point, indicating a less severe effect on enzymatic activity. In contrast, mutations in the enzymatic core (EC) (e.g., I6S, Q10P, Q10R, K48T, and H71R) resulted in a median POA production of 252 µM (Fig. 3B), which was significantly lower than the PER mutations ($P = 0.0068$). Similarly, mutations in the metal binding site (MBS) (e.g., D49N, H51R, and H57R) led to a slightly median production of 266.4 µM ($P = 0.0374$). Notably, a single mutation in the active catalytic site (ACS) (D8E) resulted in disparate POA values of 59 and 847 µM, highlighting the variability in its impact For comparison, wild-type isolates showed a median POA production of 2195 µM, suggesting that mutations in the EC and MBS regions reduced POA levels by approximately 88.5% and 88.0%, respectively, whereas mutations in PER appeared to decrease POA production by approximately 70.6%. These findings underscore the importance of mutation location in determining the extent of impact on enzymatic activity.

## Relationship between POA production and PZA susceptibility

The PZA-MIC, BACTEC Growth Index (BGI), and POA production in samples with mutations in the EC and MBS regions are shown in Table 4. In the linear regression, POA production quantified by the MODS-WQ assay was significantly associated with

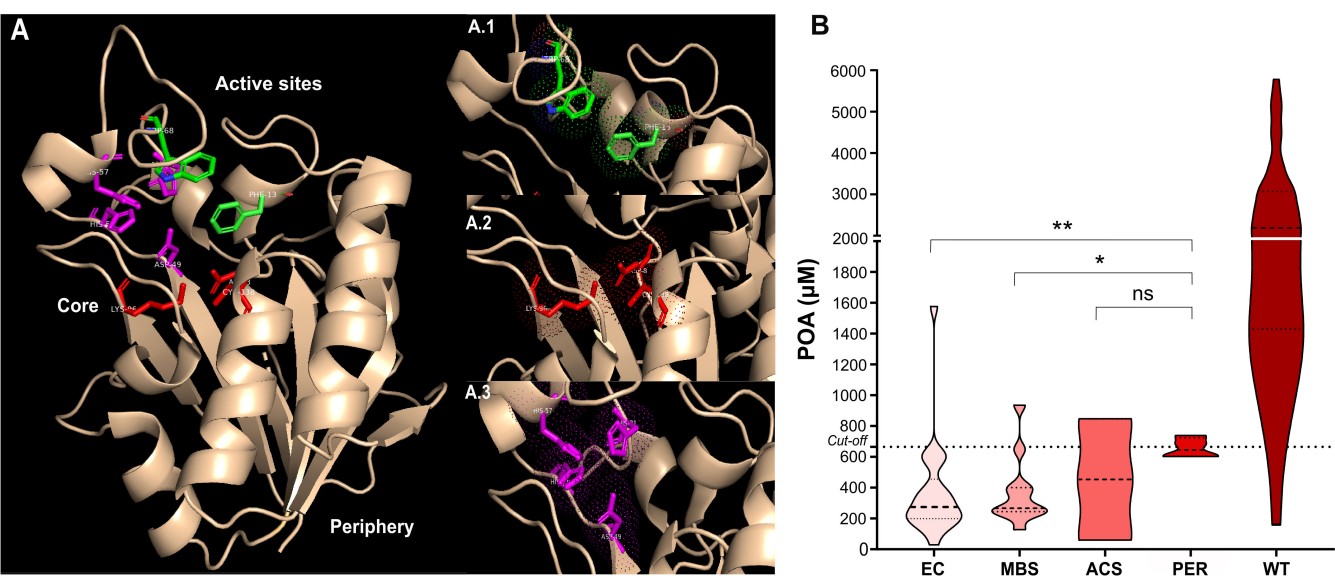

**FIG 3** Impact of pyrazinamidase mutations on pyrazinoic acid (POA) production. (A) 3-D structure of pyrazinamidase, highlighting its active sites in distinct colors for clarity: green for the PZA binding site, composed of amino acids Phe13 and Trp68 (A.1); red for the catalytic triad, consisting of Asp8, Lys96, and Cys138 (A.2); and purple for the metal-binding site (Fe+), involving Asp49, His51, and His57 (A.3). The enzyme core area adjacent to active sites and the periphery distant from these sites are also represented. (B) Violin graph presenting the distribution of POA concentrations across various mutation sites compared with wild-type (WT) samples, with the MODS-WQ$_{BC}$ assay cutoff value (665 µM) delineated by a dashed line. Categories are labeled as EC: enzymatic core, $N = 22$; MBS: metal-binding site, $N = 11$; ACS: active catalytic site, $N = 2$; periphery, $N = 4$; and WT: wild type, $N = 116$. Statistical significance is indicated by asterisks: *$P < 0.05$, **$P < 0.01$, ***$P < 0.001$, ****$P < 0.0001$, with "ns" denoting not significant, based on the Mann-Whitney test."

**TABLE 4** Minimal Inhibitory Concentration (MIC) for PZA and concentration of POA produced during culture for *M. tuberculosis* isolates with PZA mutations in the enzyme core (EC) and metal-binding site (MBS) regions

| Mutation | PZase region[a] | PZA-MIC[b] | BGI %[c] | POA production[d] |
|---|---|---|---|---|
| WT | N/A | 50 | 1% | 2346.7 ($N = 116$) |
| K48T | EC | 50 | 20% | 1466.4 ($N = 6$) |
| Q10P/Q10R | EC | 900 | 91% | 306.4 ($N = 12$) |
| D49N | MBS | 400 | 74% | 934.8 ($N = 1$) |
| H51R | MBS | 800 | 90% | 310.8 ($N = 10$) |

[a] Mutations in PZA regions, MBS: Mutations in the Metal Binding Site; EC: Mutations in the Enzymatic Core; WT: Wild-type isolate (N/A: Not applicable)
[b] PZA-MIC was measured in PZA mg/mL. PZA-MIC larger than 100 mg/mL is considered PZA resistant.
[c] BGI of less than 20% was considered an indicator of a PZA-susceptible strain.
[d] Mean of POA production in µM quantified by MODS-WQ. N represents the number of samples.

PZA resistance according to the BGI (determination coefficient $R^2 = 0.934$; $P = 0.007$) and PZA-MIC ($R^2 = 0.843$; $P = 0.028$). POA production had the highest determination coefficient for predicting PZA resistance, defined as BGI. This means that 93.4% of the variability in PZA resistance is explained by POA production for samples with localized mutations in the EC and MBS regions.

## DISCUSSION

Prompt detection of PZA resistance is crucial for effective treatment of TB. Traditional microbiological methods to evaluate PZA susceptibility face difficulties in maintaining the appropriate pH and inoculum concentrations. In our study of 264 samples, 78 (29.54%) were identified as multidrug-resistant (MDR). Among these MDR samples, 49 (62.80%) exhibited resistance to PZA, as determined using a composite reference standard. This finding aligns with other studies reporting PZA resistance rates of 38%–65% in MDR-TB cases (35–37).

The citrate buffer variant of the MODS-Wayne quantitative (MODS-WQ$_{CB}$) assay demonstrated outstanding efficacy, with a sensitivity of 92.31%, specificity of 95.93%, and a positive predictive value (PPV) of 88%. In contrast, the 7H9 matrix variant showed a notably lower performance, with a sensitivity of 81.3%, specificity of 77.2%, and PPV of 42%. These results underscore the potential of MODS-WQCB to enhance diagnostic capabilities in resource-limited settings, offering a robust tool for rapid PZA resistance detection.

Spectrophotometric measurement of POA in 7H9 media is complicated by the presence of bovine serum albumin (BSA), which is added to foster mycobacterial growth (38). BSA exhibits a higher affinity for POA than PZA at concentrations greater than 0.4%, significantly increasing the minimum inhibitory concentration (MIC)(39). Additionally, BSA tends to interact with various ligands, particularly divalent metallic cations such as copper and zinc. Notably, BSA shows a strong affinity for $Fe^{2+}$ and $Fe^{3+}$ at neutral pH, particularly near tryptophan residues, similar to lactoferrin (40–42). This affinity for POA and iron could hinder the formation of the FAS-POA complex, leading to variable sensitivity and specificity of our assay results.

The use of the citrate buffer (CB) matrix in our study not only simplified colorimetric observation but also ensured more accurate absorbance detection by minimizing medium-related biases. This contrasts with previous research by Meinzen et al. (43), which quantified POA using similar matrices but relied on samples grown on agar and stored in a culture bank. Our approach, in contrast, involves direct sampling from patient sputum. Meinzen et al. reported sensitivity and specificity rates of 96.0% and 97.4%, respectively, using a PZA concentration of 400 µg/mL, and 90.5% and 94.4% for 800 µg/mL PZA, achieving an AUC of over 93.0% for both concentrations.

Although our study did not match the high sensitivity and specificity levels reported by Meinzen et al., our approach, using direct patient samples, significantly reduced

the diagnosis time. On average, we achieved a 13.5-day reduction compared with traditional methods like the Wayne Test (30–45 days) (30) and the MODS-Wayne (mean 19.5 days). Although the performance of MODS-Wayne and our quantitative MODS-WQ were comparable (Kappa indices of 0.83 and 0.87, respectively), it is crucial to note that the variability in MODS-Wayne is mainly due to observer subjectivity. This variability could affect reproducibility (44, 45), an issue potentially resolved by our quantitative MODS-WQ variant. Further studies focusing on reproducibility are essential to validate our findings.

POA production is a key indirect marker of PZase enzymatic activity, which can be influenced by protein mutations (34, 46, 47). In our study, mutations within the PZase enzyme were categorized into four areas: ACS, MBS, EC, and PER. We observed that mutations in the EC and MBS regions led to significantly lower POA levels (252.7 µM and 266.4 µM, respectively) compared with those in PER, which produced POA levels near the 665 µM cutoff ($P < 0.01$ and $P < 0.05$, respectively). Supporting our findings, Sheen et al. (46) reported that mutations in MBS (H51R, D49N) resulted in a substantial decrease in enzymatic activity (99%) in terms of POA production, compared with the wild-type H37Rv strain. Similarly, mutations near active sites (core) led to a 72% reduction, whereas peripheral mutations caused a 47% decrease in enzymatic activity.

Our study aligns with previous research, showing a marked decrease in POA production in samples with mutations in the EC and MBS regions (88%) compared with wild-type samples. In samples with PER mutations, this reduction was slightly lower at 70.6%. Notably, our results confirmed that the POA production, in samples with mutations in EC and MBS regions, significantly explained the variability of PZA resistance according to BGI and PZA-MIC. Thus, the highly significant association (93.4%) between POA production and BGI found in this study suggests that POA production is a reliable predictor of PZA resistance in the same way as the POA efflux rate [considered the best predictor of resistance to PZA (47)].

It's important to distinguish our methodology from that of Sheen et al. (34), who directly quantified POA production in purified enzymes *in vitro*, free from external influences on the mycobacteria's POA output.

The release of POA is also influenced by other factors, like efflux pump activity. Zimic et al. (48) highlighted a strong link between POA efflux rate and PZA resistance. Their study showed that resistant isolates have lower efflux rates, whereas sensitive isolates display higher rates, allowing for the prediction of resistance with high sensitivity and specificity. By quantifying the POA in the supernatant, MODS-WQ assay captures the cumulative effect of enzymatic activity and efflux pump action, which plays a crucial role in POA transport. Our method provides a more comprehensive understanding of PZA susceptibility by accounting for both enzymatic activity and external factors, such as efflux pump activity offering a holistic view of PZA resistance mechanisms.

Enzymatic activity alone accounts for only 27.3% of PZA resistance cases (46), whereas the POA efflux rate contributes to around 51% (47). Notably, 70%–90% of PZA-resistant strains show mutations in the *pncA* gene and its promoter region (49–51). This suggests that a significant portion of resistant strains may involve other, yet unidentified, mechanisms contributing to PZA resistance.

Mutations in other genes, such as *panD* and *rpsA*, might also play a role in PZA resistance, as could the presence of heteroresistant strains with varying susceptibility genotypes (35–37, 52). Interestingly, not all mutations in the *pncA* gene are linked to PZA resistance; some result in PZA-sensitive strains (51). This complexity is illustrated in Fig. 4. Consequently, assessing POA production in the culture medium offers a more comprehensive understanding of PZA susceptibility compared with isolated enzymatic activity measurements alone.

Although we described a simple and low-cost assay to determine PZA resistance quantitatively without the necessity of primary *M. tuberculosis* isolation, this test remains unsuitable for large-scale implementation. First, there is a significant degree of sample manipulation, particularly during culture, necessitating high biosafety measures to avoid

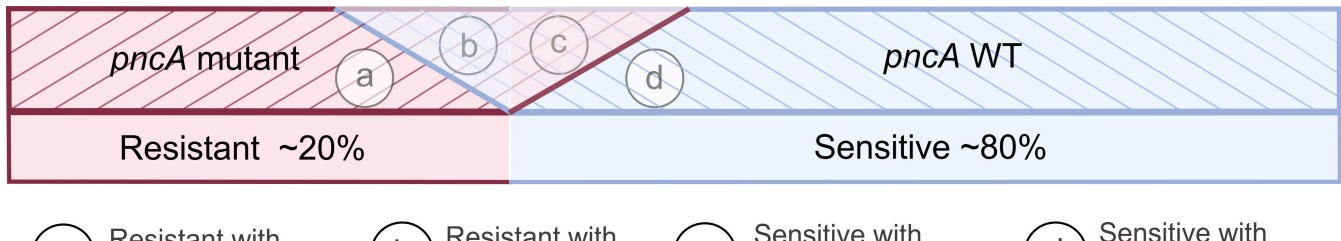

**FIG 4** Complexity of PZA susceptibility phenotype. Variability in susceptibility phenotype is a result of different factors, such as the lack of specific PZA target, the large number of potential mutations throughout the coding and promoter sequence of *pncA*, the diversity of phenotypic methods, and errors that they entail, among others. In this figure, we can observe the four types of isolates according to the susceptibility phenotype.

cross-contamination. Second, new culture media or assay buffers must be studied to improve the direct quantification of POA in the supernatant. Additionally, the results must be validated with a larger sample size.

In conclusion, the MODS-Wayne quantitative (MODS-WQ) assay, utilizing a citrate buffer, stands as an important advancement in the field of tuberculosis diagnostics, demonstrated by its exceptional predictive capacity (ROC: 97.6%, Kappa index: 0.87) and a notably swift average incubation period of merely 13 days from the collection to the culturing of sputum samples. This assay not only measures total POA production efficiently but also integrates critical diagnostic elements—specifically, the enzymatic efficiency of pyrazinamidase and the activity of efflux pumps. These components are essential for accurately determining PZA resistance, addressing one of the significant hurdles in TB treatment in regions burdened with drug-resistant strains.

Our findings are robust, showing that the POA levels quantified by the MODS-WQ assay serve as a reliable predictor of PZA resistance, particularly in samples with mutations in the EC and MBS regions. Given its high efficacy, rapid turnover, and cost-effectiveness, this method holds substantial promise for broader application, particularly in resource-limited settings where quick, reliable, and affordable diagnostic processes are crucial. This assay's design and implementation could potentially transform PZA resistance testing, making it a vital tool in global TB control strategies.

## ACKNOWLEDGMENTS

The authors extend their gratitude to the Hospital Nacional Dos de Mayo, Lima, Peru, and the Regional Tuberculosis Reference Laboratory, Callao, Lima, Peru for providing the sputum samples.

This study was funded by a Wellcome Trust Intermediate Fellowship awarded to PS.

Contributors E.T-G., M.Z., and P.S. conceived and designed the study. E.T-G. and R.A. collected clinical samples and data. E.T-G. and K.L. performed experiments and analyzed data. E.T-G. wrote the original manuscript. E.T-G. and R.A. contributed to data visualization. P.S., R.A., R.G., L.G., and M.Z. contributed to data interpretation and critically reviewed the manuscript. All authors approved the final manuscript for submission.

During the preparation of this work the author(s) used chatGPT4 and Paperpal in order to improve readability and language of the manuscript. After using this tool/service, the author(s) reviewed and edited the content as needed and take(s) full responsibility for the content of the publication.

## AUTHOR AFFILIATIONS

[1]Laboratorio de Bioinformática, Biología Molecular y Desarrollos Tecnológicos. Laboratorios de Investigación y Desarrollo. Facultad de Ciencias e Ingeniería. Universidad Peruana Cayetano Heredia, Lima, Peru

²Biomolecules Laboratory, Faculty of Health Sciences, Universidad Peruana de Ciencias Aplicadas (UPC), Lima, Peru
³Johns Hopkins Bloomberg School of Public Health, Baltimore, Maryland, USA
⁴Institute of Child Health, University College London, London, United Kingdom

## AUTHOR ORCIDs

Emily Toscano-Guerra ⓘ http://orcid.org/0000-0003-2718-1684
Patricia Sheen ⓘ http://orcid.org/0000-0002-7118-9301

## FUNDING

| Funder | Grant(s) | Author(s) |
|---|---|---|
| Grand Challenges Canada (GCC) | 0687-01-10 | Patricia Sheen |
| Wellcome Trust Intermediate | | Patricia Sheen |

## AUTHOR CONTRIBUTIONS

Emily Toscano-Guerra, Conceptualization, Data curation, Formal analysis, Investigation, Methodology, Visualization, Writing – original draft | Roberto Alcántara, Data curation, Methodology, Visualization, Writing – review and editing | Katherine Lozano Untiveros, Data curation, Methodology, Writing – review and editing | Robert Gilman, Validation, Writing – review and editing | Louis Grandjean, Data curation, Writing – review and editing | Mirko Zimic, Conceptualization, Funding acquisition, Project administration, Resources, Supervision, Validation, Writing – review and editing | Patricia Sheen, Conceptualization, Funding acquisition, Project administration, Resources, Supervision, Validation, Writing – review and editing

## DATA AVAILABILITY

The supplemental material: Supplementary methods, Supplementary Figures S1, S2, and Tables S1, S2,S3, S4 and S5 are available at DOI: 10.17632/mpf9swrznf.1. The clinical data sets used and produced in the present study are available upon reasonable request to the authors.

## ADDITIONAL FILES

The following material is available online.

Open Peer Review

**PEER REVIEW HISTORY (review-history.pdf).** An accounting of the reviewer comments and feedback.

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
