## [Reviewer comments · Microbiology Spectrum]

Microbiology Spectrum

Quantitative MODS-Wayne Assay for Rapid Detection of Pyrazinamide Resistance in *Mycobacterium tuberculosis* from Sputum Samples

Emily Toscano Guerra, Roberto Alcántara, Katherin Lozano, Robert Gilman, Louis Grandjean, Mirko Zimic, and Patricia Sheen

Corresponding Author(s): Emily Toscano Guerra, Universidad Peruana Cayetano Heredia

Review Timeline:

Submission Date:	February 19, 2024
Editorial Decision:	July 1, 2024
Revision Received:	August 23, 2024
Accepted:	October 12, 2024

Editor: Fei Chen

Reviewer(s): Disclosure of reviewer identity is with reference to reviewer comments included in decision letter(s). The following individuals involved in review of your submission have agreed to reveal their identity: Xia Yu (Reviewer #1)

Transaction Report:

DOI: <https://doi.org/10.1128/spectrum.00471-24>

Re: Spectrum00471-24 (Quantitative MODS-Wayne Assay for Rapid Detection of Pyrazinamide Resistance in *Mycobacterium tuberculosis* from Sputum Samples)

Dear Dr. Emily Toscano Guerra:

Thank you for the privilege of reviewing your work. Below you will find my comments, instructions from the Spectrum editorial office, and the reviewer comments.

Revision Guidelines

Sincerely,
Fei Chen
Editor
Microbiology Spectrum

Reviewer #1 (Comments for the Author):

See attach file.

Reviewer #2 (Comments for the Author):

The manuscript aims to refine the MODS-Wayne test by introducing a quantitative approach (MODS-WQ) using sputum samples from patients with TB, and seeks to reduce the diagnosis time, eliminate the need for pre-isolation in solid cultures, and move beyond qualitative judgments. While straightforward, there are areas for improvement to enhance readability and comprehension:

The MODS-Wayne Assay was used to detect Pyrazinamide Resistance in Mycobacterium tuberculosis isolates, not directly from Sputum Samples, the title needs to be changed.

What is the principle of Microscopic Observation Drug Susceptibility (MODS) assay? And the advantages compared to liquid and solid drug susceptibility test? Clarity and Elaboration in the introduction.

Abbreviations should be consistent throughout the manuscript, such as pyrazinamidase (PZase) was shown in line 15 of page 3, but pyrazinamidase (PZAse) in line 2 of page 4. After the full name and abbreviation are used for the first time, either can be used in the future. Line 21 in page 8, MP and TBN should be supplemented the full name.

How the 264 TB-positive were identified? Reference for a final concentration INH and RIF should be cited.

As shown in Table1, MP (169) represents samples collected in the study 2015-2016; TBN (95) represents samples collected in the study 2017-2018. But why 169 samples conducted by MODS-WQCB, and 95 samples conducted by MODS-WQ7H9? What is the basis? Besides, why choose citrate buffer matrix?

Description the limitations of this study.

Toscano-Guerra et al report the performance of MODS-Wayne Assay for Rapid Detection of Pyrazinamide Resistance in Mycobacterium tuberculosis from Sputum Samples. Overall, this is an interesting body of work that provides new approach for PZA susceptible test.

Major comments:

1. The authors propose that MODS-WQ approach accounts for critical factors influencing PZA resistance, including efflux pump activity, the data supporting this conclusion are not provided.
2. Since the sample size of both MP and TBN group are not sufficient large, combination of these two group for evaluating MODS-Wayne assay should be added.
3. According to the WHO catalog of tuberculosis mutations associated with drug resistance and SuspectPZA web tool, mutations detected in this study was categorized as PZA resistant or susceptible. The POA production by PZA resistant or susceptible mutations should be analyze.

Minor comments

- 1.Line5,P6. Upon growth in the PZA-C well, does this observe by naked eyes or microscopy?
- 2.Table 1 needs statistical analysis
- 3.Table 2 is not necessary and remove it.
4. Please provide the PDB code of pyrazinamidase.

The manuscript aims to refine the MODS-Wayne test by introducing a quantitative approach (MODS-WQ) using sputum samples from patients with TB, and seeks to reduce the diagnosis time, eliminate the need for pre-isolation in solid cultures, and move beyond qualitative judgments. While straightforward, there are areas for improvement to enhance readability and comprehension:

The MODS-Wayne Assay was used to detect Pyrazinamide Resistance in *Mycobacterium tuberculosis* isolates, not directly from Sputum Samples, the title needs to be changed.

What is the principle of Microscopic Observation Drug Susceptibility (MODS) assay? And the advantages compared to liquid and solid drug susceptibility test? Clarity and Elaboration in the introduction.

Abbreviations should be consistent throughout the manuscript, such as pyrazinamidase (PZase) was shown in line 15 of page 3, but pyrazinamidase (PZAse) in line 2 of page 4. After the full name and abbreviation are used for the first time, either can be used in the future. Line 21 in page 8, MP and TBN should be supplemented the full name.

How the 264 TB-positive were identified? Reference for a final concentration INH and RIF should be cited.

As shown in Table1, MP (169) represents samples collected in the study 2015-2016; TBN (95) represents samples collected in the study 2017-2018. But why 169 samples conducted by MODS-WQCB, and 95 samples conducted by MODS-WQ7H9? What is the basis? Besides, why choose citrate buffer matrix?

Description the limitations of this study.

Reviewer 1.

Toscano-Guerra et al report the performance of MODS-Wayne Assay for Rapid Detection of Pyrazinamide Resistance in Mycobacterium tuberculosis from Sputum Samples. Overall, this is an interesting body of work that provides new approach for PZA susceptible test.

Authors: We would like to thank the reviewer for their positive feedback and for providing the opportunity to clarify and enhance our manuscript. We believe that their comments have helped us to further refine and improve the presentation of our findings.

Major comments:

Comment 1. The authors propose that MODS-WQ approach accounts for critical factors influencing PZA resistance, including efflux pump activity, the data supporting this conclusion are not provided.

Answer 1: Thank you for highlighting this point. In the context of PZA resistance, the MODS-WQ assay measures the total Pyrazinoic Acid (POA) produced during the incubation and that is present in the supernatant. POA is the active metabolite of Pyrazinamide (PZA), formed by the enzymatic action of pyrazinamidase. In resistant strains, PZA is not effectively converted to POA or is expelled via efflux pumps, impacting the observed resistance profile, as indicating by several previous reports already cited in the manuscript (Zimic, Fuentes, et al. 2012; Zimic, Loli, et al. 2012). The POA detected in the medium reflects the combined effects of pyrazinamidase activity and efflux pump function, as it represents the amount both produced and expelled by the bacteria. Therefore, while we did not present specific data about efflux pump or enzyme activity, the measurement of POA in the supernatant inherently includes the contributions of both pyrazinamidase enzyme activity and

efflux pump function. This measurement provides an indirect but comprehensive view of the factors influencing PZA resistance. We have clarified this aspect in the manuscript to better communicate how the MODS-WQ assay accounts for efflux pump activity. We appreciate your feedback and will ensure this will be more explicitly explained in the discussion of the “marked-up manuscript” (page 14, line 15 – 20) and conclusion (page 15, line 11 – 13)

Comment 2. Since the sample size of both MP and TBN group are not sufficient large, combination of these two group for evaluating MODS-Wayne assay should be added.

Answer 2: Thank you for your suggestion. We have addressed this by combining the MP and TBN groups for a more comprehensive evaluation of the MODS-Wayne assay performance (Page 9, line 27 – 31) and new Table 3.

Comment 3. According to the WHO catalog of tuberculosis mutations associated with drug resistance and SuspectPZA web tool, mutations detected in this study was categorized as PZA resistant or susceptible. The POA production by PZA resistant or susceptible mutations should be analyze.

Answer 3: Thank you for your recommendation. We have conducted the requested analysis and included a new table (supplementary table S5) that presents the POA production associated with each individual mutation. This complements our previous analysis by providing a more detailed view of the impact of each specific mutation (Page 10, line 26 – Page 11, line 2).

Minor comments

- 1.) Line5,P6. Upon growth in the PZA-C well, does this observe by naked

eyes or microscopy?

Answer: Thanks for the opportunity to clarify this point. PZA-C well should be evaluated by microscope in order to ensure the strains growth before evaluating the resistance profile. We have mentioned this in the marked-up manuscript (Page 6, line 12)

2) Table 1 needs statistical analysis

Answer: As you suggested, we have added statistical analysis to this table.

3) Table 2 is not necessary and remove it.

Answer: As you suggested, we have removed it from the manuscript and moved to supplementary files.

4) Please provide the PDB code of pyrazinamidase.

Answer: As you suggested we have added the PDB code: 3PL1, in methods section (Page 8, line 12)

Reviewer 2:

The manuscript aims to refine the MODS-Wayne test by introducing a quantitative approach (MODS-WQ) using sputum samples from patients with TB, and seeks to reduce the diagnosis time, eliminate the need for pre-isolation in solid cultures, and move beyond qualitative judgments. While straightforward, there are areas for improvement to enhance readability and comprehension.

Authors: We would like to express our gratitude to the reviewer for their meticulous and insightful review. We believe that their feedback has significantly improved the quality of our manuscript.

Comment 1. The MODS-Wayne Assay was used to detect Pyrazinamide Resistance in *Mycobacterium tuberculosis* isolates, not directly from Sputum Samples, the title needs to be changed

Answer 1: Thank you for your feedback. We appreciate the opportunity to clarify the methodology used in our study. Our method involves the following steps:

1. **Direct Processing:** We begin with raw sputum samples collected from patients, ensuring that our approach eliminates the need for initial isolation on solid media.
2. **Sample Preparation:** The sputum samples are decontaminated and concentrated using standard procedures, which allow us to work directly with them for the MODS-WQ assay.
3. **Cultivation and Detection:** We directly inoculate these prepared sputum samples into the liquid culture system used for the MODS-WQ assay. This allows us to rapidly detect pyrazinamide resistance without the intermediate steps typically required in traditional methods.

By implementing these steps, we streamline the process, reduce diagnosis time, and maintain accuracy in detecting drug resistance. We believe that the current title reflects these innovations and highlights the significance of our approach in advancing TB diagnostics, as we directly utilize sputum samples rather than relying on pre-isolated *Mycobacterium tuberculosis* strains.

Comment 2. What is the principle of Microscopic Observation Drug Susceptibility (MODS) assay? And the advantages compared to liquid and solid drug susceptibility test? Clarity and Elaboration in the introduction.

Answer 2: Thank you for your comment. We have revised the introduction to include a detailed explanation of the principle of the Microscopic Observation Drug Susceptibility (MODS) assay in the “marked-up manuscript” (Page 4, line 16 – 28). The MODS assay is based on the microscopic detection of characteristic cord-like growth patterns of *Mycobacterium tuberculosis* in liquid culture, in the presence or absence of drugs. This assay allows for the rapid identification of drug susceptibility by observing the growth inhibition of the bacteria.

Compared to traditional liquid and solid drug susceptibility tests, the MODS assay offers several advantages. It is faster, providing results within 5 to 21 days, whereas solid culture methods can take up to 8 weeks. It is also cost-effective and less labor-intensive, making it suitable for resource-limited settings. Additionally, the MODS assay has high sensitivity and specificity, making it an effective tool for early detection and management of drug-resistant tuberculosis.

Comment 3. Abbreviations should be consistent throughout the manuscript, such as pyrazinamidase (PZase) was shown in line 15 of page 3, but pyrazinamidase (PZAse) in line 2 of page 4. After the full name and abbreviation are used for the first time, either can be used in the future.

Answer 3: Thanks for the correction. It has been modified.

Comment 4. Line 21 in page 8, MP and TBN should be supplemented the full name.

Answer 4: MP and TBN are not abbreviations of full names describing any particular sample characteristics. They are only random assigned codes regarding the sample origin collection: Hospital Nacional Dos de Mayo (MP) OR the Regional Tuberculosis Reference Laboratory (TBN), both located in Lima, Peru, as indicated in the manuscript.

Comment 5. How the 264 TB-positive were identified? Reference for a final concentration INH and RIF should be cited.

Answer 5: Thank you for your question. The 264 TB-positive samples were identified using the acid-fast bacilli smear microscopy method, which is a widely used technique for detecting *Mycobacterium tuberculosis* in sputum samples. This point has been explained in the method section for better clarification (Page 5, line 17 – 18). In addition, a reference for the final concentration of INH and RIF have been properly added in the marked-up manuscript (Page 6, line 5).

Comment 6. As shown in Table1, MP (169) represents samples collected in the study 2015-2016; TBN (95) represents samples collected in the study 2017- 2018. But why 169 samples conducted by MODS-WQCB, and 95 samples conducted by MODS-WQ7H9? What is the basis? Besides, why choose citrate buffer matrix?

Answer 6: Since we used remaining store sputum samples for the study, the number of samples used for each approach was based principally on the availability of samples and resources at the time of the study. There was no statistical sample size calculation. However, we decided to use a higher number of samples for the MODS-WQ_{BC} (MP) since this was our novel approach that involves more steps in sample processing to detect POA in comparison to the MODS-WQ_{7H9} (TBN) that was more similar to the reported MODS-Wayne assay (Alcántara et al. 2019). We aimed to evaluate how different methodological approaches could impact the detection of Pyrazinamide resistance. This comparative analysis allowed us to assess the strengths and limitations of each method, providing valuable insights into their efficacy and practicality in diverse contexts.

On the other hand, the use of citrate buffer is based on previous studies of our research group where we use this buffer for POA detection (Meinzen et al. 2016). In those cases, we did not observe any interference with POA detection and the SAF

reagent used for the colorimetric output.

Comment 7. Description the limitations of this study.

Answer 7: Thank you for your suggestion. As you recommended, we have added the limitations of our study in the discussion section, before the conclusion (Page 15, line 1 – 7). We have elaborated on these points in the manuscript to provide a comprehensive understanding of the study's limitations. We appreciate your feedback and believe that this addition strengthens the overall quality of our paper.

Re: Spectrum00471-24R1 (Quantitative MODS-Wayne Assay for Rapid Detection of Pyrazinamide Resistance in *Mycobacterium tuberculosis* from Sputum Samples)

Dear Dr. Emily Toscano Guerra:

Your manuscript has been accepted, and I am forwarding it to the ASM production staff for publication. Your paper will first be checked to make sure all elements meet the technical requirements. ASM staff will contact you if anything needs to be revised before copyediting and production can begin. Otherwise, you will be notified when your proofs are ready to be viewed.

Sincerely,
Fei Chen
Editor
Microbiology Spectrum

Reviewer #1 (Comments for the Author):

I suggest this manuscript could be accepted.